

# The complete mitochondrial genome of *Pyxicephalus adspersus*: high gene rearrangement and phylogenetics of one of the world's largest frogs

Yin-Yin Cai[1], Shi-Qi Shen[1], Li-Xu Lu[1], Kenneth B. Storey[2], Dan-Na Yu[1,3] and Jia-Yong Zhang[1,3]

[1] College of Chemistry and Life Science, Zhejiang Normal University, Jinhua, Zhejiang, China
[2] Department of Biology, Carleton University, Ottawa, ON, Canada
[3] Key Lab of Wildlife Biotechnology, Conservation and Utilization of Zhejiang Province, Zhejiang Normal University, Jinhua, Zhejiang, China

## ABSTRACT

The family Pyxicephalidae including two subfamilies (Cacosterninae and Pyxicephalinae) is an ecologically important group of frogs distributed in sub-Saharan Africa. However, its phylogenetic position among the Anura has remained uncertain. The present study determined the complete mitochondrial genome sequence of *Pyxicephalus adspersus*, the first representative mitochondrial genome from the Pyxicephalinae, and reconstructed the phylogenetic relationships within Ranoidae using 10 mitochondrial protein-coding genes of 59 frog species. The *P. adspersus* mitochondrial genome showed major gene rearrangement and an exceptionally long length that is not shared with other Ranoidae species. The genome is 24,317 bp in length, and contains 15 protein-coding genes (including extra *COX3* and *Cyt b* genes), four rRNA genes (including extra *12S rRNA* and *16S rRNA* genes), 29 tRNA genes (including extra $tRNA^{Leu\ (UAG)}$, $tRNA^{Leu\ (UUR)}$, $tRNA^{Thr}$, $tRNA^{Pro}$, $tRNA^{Phe}$, $tRNA^{Val}$, $tRNA^{Gln}$ genes) and two control regions (CRs). The Dimer-Mitogenome and Tandem duplication and random loss models were used to explain these gene arrangements. Finally, both Bayesian inference and maximum likelihood analyses supported the conclusion that Pyxicephalidae was monophyletic and that Pyxicephalidae was the sister clade of (Petropedetidae + Ptychadenidae).

## BACKGROUND

Anuran mitochondrial (mt) genomes are closed double-stranded DNA molecules with the lengths varying from 15 to 23 kb (*Kakehashi et al., 2013*; *Zhang et al., 2018a*) and typically encode 37 genes: two ribosomal RNAs (12S and 16S rRNAs), 22 transfer RNAs (tRNAs), 13 protein-coding genes (PCGs), and the control region (CR) or D-loop region (*Boore, 1999*, *2000*). The mt genome has several valuable characteristics that have led to its wide use as a research tool including limited recombination, maternal inheritance, rapid evolutionary rate, small size, and relatively conserved gene arrangement (*Avise, 1994*). Complete or partial mt genomes have been extensively used to test evolutionary

Corresponding authors
Dan-Na Yu, ydn@zjnu.cn
Jia-Yong Zhang,
zhangjiayong@zjnu.cn

hypotheses, to study population biogeography and phylogenetic relationships, and to distinguish cryptic species (*Boore, 1999*; *Yu et al., 2015*; *Zhang et al., 2008*).

To date, there are 7,977 known species of Anura (*Frost, 2019*) on Earth, the vast majority of which have not had their mt genomes sequenced and studied. According to *Jacob Machado et al. (2018)* and *Zhang et al. (2018a)*, there are partial mt genomes for over one hundred species representing 35 families (72 genera) and complete mt genomes for over two hundred species representing 27 families (76 genera) listed in GenBank. Although non-neobatrachian frogs show conservation of the typical mt gene organization, ranoid frogs belonging to the neobatrachia show a variety of gene rearrangements in their mt genomes (*Kurabayashi & Sumida, 2013*; *Zhang et al., 2013*). Hence, by increasing the number of sequenced frog mt genomes, we can gain valuable information about mt genome arrangements and improve our understanding of mt genomic phylogenetics and evolution, population genetics (*Cai et al., 2018*; *Cheng et al., 2018*; *Lin et al., 2014*; *Ni et al., 2015*; *Ye et al., 2016*) and mt genome expression (*Zhang et al., 2019*).

Some gene rearrangements in the mt genome have been reported in neobatrachians. For example, some species show a rearrangement of the *ND5* gene (*Alam et al., 2010*; *Anoop et al., 2017*; *Kumar et al., 2017*; *Li et al., 2014a*, *2016a*; *Sano et al., 2004*, *2005*). A gene rearrangement of *ND6* gene has also been reported in several species (*Zhang et al., 2018a*; *Irisarri et al., 2010*; *Carr et al., 2015*; *Kakehashi et al., 2013*). Two $tRNA^{Met}$ genes have also been found in many species of Ceratobatrachidae, Dicroglossidae, Mantoidae, and Megophryidae (*Jiang et al., 2018*; *Kiran et al., 2017*; *Kurabayashi et al., 2008*; *Li et al., 2018a*, *2018b*; *Liang et al., 2016*). *Mantella madagascariensis* (*Kurabayashi et al., 2006*), *Rana kunyuensis* (*Li et al., 2016a*), and *Rhacophorus schlegelii* (*Sano et al., 2005*) possess duplicated CRs. By contrast, the $tRNA^{Thr}$ gene was lost in *Nanorana taihangnica* (*Chen et al., 2015*) and the *ATP8* and *ND5* genes were lost in *Polypedates megacephalus* (*Zhang et al., 2005*).

The phylogenetic relationship of Pyxicephalidae was hotly disputed. Pyxicephalidae contains two subfamilies (Cacosterninae and Pyxicephalinae) with a total of 12 genera and 84 recognized species according to molecular and morphological data (*Frost et al., 2006*). Using mtDNA and morphological data, Van Der Meijden suggested that Pyxicephalinae, which is composed solely of *Aubria* and *Pyxicephalus* (*Frost et al., 2006*), was imbedded within the Discroglossine (*Van Der Meijden et al., 2005*). By contrast, *Wiens et al. (2009)* considered Pyxicephalinae as one subfamily of Ranidae. *Frost et al. (2006)* and *Roelants et al. (2007)* recovered the monophyly of Pyxicephalidae and *Van Der Meijden et al. (2011)* and *Pyron & Wiens (2011)* treated Pyxicephalidae as a valid family. *Zhang et al. (2013)* supported Pyxicephalidae as a sister clade to Petropedetidae. *Bittencourt-Silva et al. (2016)* discussed the phylogenetic relationship of the genera in Pyxicephalidae and concluded that *Aubria* and *Pyxicephalus*, belonging to Pyxicephalinae, were the basal clade of Pyxicephalidae. Hence, research on the phylogenetic relationships of *Pyxicephalus* could reveal the monophyly of Pyxicephalidae.

The African giant bullfrog (*Pyxicephalus adspersus*) belongs to the subfamily Pyxicephalinae of the family Pyxicephalidae. It is an explosive-breeding anuran that inhabits arid to subtropical grasslands and savanna across most of southern Africa

 

(*Channing, 2001*; *Minter et al., 2004*). It is one of the world's largest amphibians, with males (unusually larger than females) reaching a length of 225 mm (*Wager, 1965*) and weights of 1.4–2.0 kg (*Channing, DU Preez & Passmore, 1994*; *Loveridge, 1950*). *Liu et al. (2013)* found *Scapharca broughtonii* (Mollusca: Bivalvia), which is an enlarged species compared to other species of *Scapharca*, had largest mitochondrial genomes among the genus *Scapharca* (*Liu et al., 2013*). A species with a giant size within an animal group may be associated with a larger mitogenome (*Alam et al., 2010*; *Yu et al., 2012b*, *2015*; *Zhang et al., 2018a*). However, no studies in *Pyxicephalus adspersus* has examined the potential involvement of mt genome evolution. So we wondered whether this giant bullfrog had novel features of its mt genome. In the present study, we sequenced the mt genome of *Pyxicephalus adspersus*, the first representative mt genome of the Pyxicephalinae, to determine if mitochondrial gene rearrangement occurred and to study the phylogenetic relationships of Pyxicephalidae.

## MATERIALS AND METHODS

### Animal treatments

*Pyxicephalus adspersus* are not protected by provisions in the laws of the People's Republic of China on the protection of wildlife. Two samples of *Pyxicephalus adspersus* were purchased from a pet market and bred in the laboratory of JY Zhang at the College of Life Science, Zhejiang Normal University. Sample acquisition was reviewed, approved and carried out in accordance with the relevant guidelines of the Committee of Animal Research Ethics of Zhejiang Normal University.

### DNA extraction, PCR and sequencing

Total DNA was extracted from the clipped toe of one specimen using a DNeasy Tissue Kit (Qiagen, Hilden, Germany). We amplified 13 overlapping gene fragments by normal polymerase chain reaction (PCR) and long-and-accurate (LA) PCR methods slightly modified from *Yu et al. (2015)* and *Zhang et al. (2013)*. However, these methods failed to find some mitochondrial genes including *ND3*, *ND5*, and *tRNA^{Ile}*, *tRNA^{Cys}*, *tRNA^{Lys}*, and *tRNA^{Arg}*. We subsequently designed one pair of primers to sequence the *ND5* gene (ND5-J: ATRGARGGNCCNACACCWGT; ND5-N: CCCATNTTWCGRATRTCCTGGTC) based to the known mitochondrial gene sequences of 53 Ranidae species from GenBank. After we obtained the *ND5* gene fragment, we then designed two pairs of specific primers for *ND6* (ND65-J: ACAAGAGCAGAACAATAAGC, ND65-N: TAGAGTGGAGTAAGGCAGAA and ND56-J: ATACAACCGAATTGGAGACA, ND56-N: GGTAAATCAGTGGGTAGGTAT) to sequence the fragment ranging from *ND6* to *ND5* genes and the fragment ranging from *ND5* to *ND6* genes, respectively. Surprisingly, when the two fragments from *ND6* to *ND5* and *ND5* to *ND6* were amplified their sizes proved to be nearly 8,000 and 15,000 bp, respectively. All PCR was performed using a programable Thermal Cycler (Veriti PCR Thermal Cycler; Applied Biosystems, Foster City, CA, USA) or a MyCycler Thermal Cycler (Bio-Rad, Hercules, CA, USA). *TaKaRa LA-Taq kits* and *TaKaRa Ex-Taq* (Takara Biomedical, Dalian, China) were used for LA-PCR and normal PCR reactions, respectively. PCR products were electrophoresed

on 1% or 2% agarose gels and sequences were obtained using an ABI 3730 automated DNA sequencer (Applied Biosystems, Foster City, CA, USA) using the primers walking method for both strands.

## Sequence assembly, annotation and analysis

Sequences were checked and assembled using SeqMan (Lasergene version 5.0) (*Burland, 2000*). All mt genes were determined by Mitos WebServer (*Bernt et al., 2013*). The locations of the 15 protein coding genes and four rRNA genes were further identified by comparison with the homologous sequences of closely related anurans downloaded from GenBank. All tRNA genes were further determined using tRNA-scan SE 2.0 (*Chan & Lowe, 2016*) or determined by comparison with the homologous sequences of other anurans. The mt genome was deposited in GenBank with accession number MK460224. The mt genome map of *Pyxicephalus adspersus* was formed using GenomeVx (*Conant & Wolfe, 2008*) (http://wolfe.ucd.ie/GenomeVx/). The A+T content, codon usage and relative synonymous codon usage (RSCU) of PCGs were calculated by Mega 7.0 (*Kumar, Stecher & Tamura, 2016*). Composition skewness was calculated according to the following formulae: AT-skew = (A−T)/(A+T); GC-skew = (G−C)/(G+C) (*Perna & Kocher, 1995*).

## Molecular phylogenetic analysis

The phylogenetic analyses was performed using the combined nucleotide datasets by the Bayesian inference (BI) and maximum likelihood (ML) methods. BI and ML analyses were performed with 59 anuran mt genomes including *Pyxicephalus adspersus*. In total this included 55 species as the ingroup from Ranidae, Dicroglossidae, Rhacophoridae, Mantellidae, Pyxicephalidae, Petropedetidae, Ptychadenidae, Ceratobatrachidae, and Phrynobatrachidae (*Alam et al., 2010*; *Chen et al., 2011*; *Hofman et al., 2012*; *Huang et al., 2016a*, *2016b*; *Jiang et al., 2017*; *Kakehashi et al., 2013*; *Kurabayashi et al., 2006*, *2010*; *Li et al., 2014b*, *2016b*; *Lin et al., 2014*; *Liu, Wang & Zhao, 2017a*; *Liu et al., 2017b*; *Liu, Wang & Su, 2005*; *Ni et al., 2015*; *Ren et al., 2009*; *Sano et al., 2004*, *2005*; *Xia et al., 2014*; *Yan et al., 2016*; *Yang et al., 2018*; *Yu, Zhang & Zheng, 2012a*; *Yu et al., 2012b*, *2015*; *Zhang, Xia & Zeng, 2016*; *Zhang et al., 2005*, *2009*, *2013*, *2018a*; *Zhao, Meng & Su, 2018*; *Zhou et al., 2009*) and four species as outgroups from Microhylidae (*Chen et al., 2016*; *Wang et al., 2018*; *Zhao, Meng & Su, 2018*). We used the nucleotide data to assess BI and ML topology to discuss the phylogenetic position of *Pyxicephalus*. Although extra *COX3* and *Cyt b* genes were found in *Pyxicephalus adspersus* (see the result in the following context), the extra copies of *COX3* and *Cyt b* genes were identical to the other *COX3* and *Cyt b* genes (100% similarity). Therefore, in phylogenetic analyses, only one set genes of *COX3* and *Cytb* was used. In addition, due to some mitochondrial PCGs missing in some species (*ND5*), lacking good information (*ATP8*) and the heterogeneous base composition and poor phylogenetic performance *(ND6)* (*Zhang et al., 2018a*), 10 PCGs genes were used in this study and separately aligned in Mega 7.0 (*Kumar, Stecher & Tamura, 2016*). All genes were split jointed, clustered, Gblocked and concatenated using PhyloSuite v1.1.13 (*Zhang et al., 2018b*). An alignment of the 10 mt PCGs dataset consisting of 7,244 nucleotides sites was concatenated. To obtain the substitution model of

the 10 mt PCGs dataset, data partitioning schemes were compared according to the Bayesian information criterion using the program PartitionFinder v1.0 (*Lanfear et al., 2012*). We set the 10 PCGs as 30 partitions in the 10 mt PCGs dataset according to the codon positions (the first, second, and third position) and gene numbers (10 PCGs). The best substitution model of the 30 partitions of the 10 mt PCGs dataset is shown in Table 1. Next, a GTRGAMMAI model in the RaxML program (*Stamatakis, 2006*) for the 10 mt PCGs dataset with 30 partitions was used for ML analysis and the GTR+I+G model in the MrBayes3.1.2 (*Huelsenbeck & Ronquist, 2001*) was used for Bayesian analysis. During BI analysis, the following settings were applied: number of Markov chain Monte Carlo generations = 10 million; sampling frequency = 1,000; burn-in = 1,000. The burn–in size was determined by checking convergences of -log likelihood. Bayesian runs achieved sufficient convergence when the average standard deviation of split frequencies was below 0.01.

# RESULTS AND DISCUSSION

## Mitogenome characteristics of *Pyxicephalus adspersus*

The length of the *Pyxicephalus adspersus* mt genome was 24,317 bp and the mt genome encoded 15 PCGs (including extra *COX3* and *Cyt b* genes), four rRNA genes (two each of *12S rRNA* and *16S rRNA* genes), 29 tRNA genes (including extra $tRNA^{Leu\,(UAG)}$, $tRNA^{Thr}$, $tRNA^{Pro}$, $tRNA^{Phe}$, $tRNA^{Leu\,(UUR)}$, $tRNA^{Val}$, and $tRNA^{Gln}$ genes) and two CRs (including one extra CR) (Fig. 1; Table 2). All PCGs excluding the *ND6* gene and four rRNA genes as well as all tRNA genes excluding $tRNA^{Ala}$, $tRNA^{Asn}$, $tRNA^{Cys}$, $tRNA^{Gln}$, $tRNA^{Glu}$, $tRNA^{Pro}$, $tRNA^{Ser}$, $tRNA^{Tyr}$ genes were encoded on the major strand. The length of the *Pyxicephalus adspersus* mitogenome was the largest size among all known anuran mitogenomes.

A total 22 bp intergenic overlap inferring to 10 genes was found in the mt genome of *Pyxicephalus adspersus*. The total non-coding regions (NCRs) in the mitogenome was 1,376 bp, composed of 13 larger NCRs ranging from 22 to 324 bp and many smaller regions (1–10 bp) (Fig. 1; Table 2). NCR13 between *ND5* and *Glu* was 324 bp and showed 92.6% similarity with the *ND6* gene. NCR11 between *COX3* and *ND3* genes showed 73.9% similarity with the $tRNA^{Gly}$ gene and NCR7 between $tRNA^{Leu}$ and $tRNA^{Ile}$ genes was 100% similar to the 5′ segment of the *ND1* gene. The other spacer regions showed 48.7–67.3% similarity with corresponding gene clusters (Table 3).

A total of 13 NCRs also had high homology compared to corresponding deleted genes (Table 3). The $tRNA^{Ile}$ gene moved from the typical *IQM* tRNA cluster to a position between the 22 bp NCR7 and a 219 bp NCR8 and the corresponding location of the $tRNA^{Ile}$ gene between the *ND1* and $tRNA^{Gln}$ genes was replaced by a 60 bp NCR1 (50% similarity with *Ile*). The $O_L$-$tRNA^{Cys}$ genes also moved from the $WANO_LCY$ tRNA cluster to a position between a 219 bp NCR8 and the $tRNA^{Asp}$ gene, whereas a 39 bp NCR2 was formed in the position between $tRNA^{Asn}$ and $tRNA^{Tyr}$ genes. The $tRNA^{Asp}$ gene moved from the typical position between $tRNA^{Ser}$ and *COX2* genes to a location between the $tRNA^{Cys}$ gene and a 151 bp NCR9 and the former position was replaced by a 59 bp NCR3. The $tRNA^{Lys}$ gene moved from the typical position between *COX2* and *ATP8* genes to a

**Table 1 The partition schemes and best-fitting models selected of 10 protein-coding genes in the PCG123 data.**

| Subset | Subset partitions | Best model |
| --- | --- | --- |
| Partition 1 | atp6_pos1, cox3_pos3, nd1_pos1, nd2_pos1, nd3_pos1, nd4_pos1 | GTR+I+G |
| Partition 2 | atp6_pos2, cox2_pos2, nd2_pos2, nd3_pos2, nd4_pos2, nd4l_pos2 | TVM+I+G |
| Partition 3 | atp6_pos3, nd1_pos3, nd2_pos3, nd3_pos3, nd4_pos3, nd4l_pos3 | TrN+G |
| Partition 4 | cox1_pos1, cox3_pos1 | SYM+I+G |
| Partition 5 | cox1_pos2, cytb_pos2, nd1_pos2 | TIM+I+G |
| Partition 6 | cox1_pos3 | TrN+G |
| Partition 7 | cox2_pos1, cytb_pos1, nd4l_pos1 | SYM+I+G |
| Partition 8 | cox2_pos3 | TrN+G |
| Partition 9 | cox3_pos2 | JC |
| Partition 10 | cytb_pos3 | TrN+I+G |

location between a 151 bp NCR9 and a 54 bp NCR10 and the former position was replaced by a 61 bp NCR3. The *ND3-tRNA$^{Arg}$* genes moved from the typical position between *tRNA$^{Gly}$* and *ND4L* genes to a location between a 65 NCR11 and ND5 and the former position was replaced by a 150 bp NCR5. The *ND5* gene moved from the typical position between *tRNA$^{Ser}$* and *ND6* genes to a location between the 98 bp NCR12 and the 324 bp NCR13 and the former position was replaced by a 143 bp NCR6.

There were 15 PCGs in the mt genome of *Pyxicephalus adspersus*, including two *COX1* and two *Cyt b* genes; in both cases the two copies of each gene were identical. PCGs started with ATN codons except for *COX1*, which used GTG. Most PCGs ended with a complete TAN codon or with an incomplete T or TA except for *ND5* and *ND6* that showed AGG as the termination codon. RSCU and codon counts for the mt genome of *Pyxicephalus adspersus* are shown in Fig. 2. It was evident that the most frequently used codon was CGA, followed by GCC, CAA, and CCC. Leu (UUR), Ile, and Phe as the most frequently used amino acids were also found.

The genome composition (A: 29.5%, C: 27.8%, G: 14.5%, T: 28.2%) showed an A+T bias, which accounted for 57.7% of the bases, and exhibited a negative GC-skew (−0.31) and a positive AT-skew (0.02) (Table 4). The highest A+T content (65.5%) was found in the CR, whereas the lowest A+T content (56.5%) was found in the PCGs region.

Two copies of the gene cluster (*tRNA$^{Glu}$-Cyt b-CR-tRNA$^{Leu}$-tRNA$^{Thr}$-tRNA$^{Pro}$-tRNA$^{Phe}$-12S RNA-tRNA$^{Val}$-16S RNA-tRNA$^{Leu}$*) occurred, one between *ND6* and *tRNA$^{Ile}$* genes and the other between *ND5* and *ND1* genes. Remarkably, the nearly identical nucleotide sequences (99.99% similarity with only one G base extra inserted into *tRNA$^{Glu}$* of the 5,454 alignment sites) were found in the two copies of these gene clusters The extra *COX3* gene was located between *tRNA$^{Lys}$* and *ND3* genes. The *tRNA$^{Ile}$* gene moved from the typical neobatrachian IQM tRNA cluster to a location between the *tRNA$^{Leu}$* and *tRNA$^{Cys}$* genes whereas the *tRNA$^{Cys}$* gene moved from the typical neobatrachian WANCY tRNA cluster to a location between the *tRNA$^{Ile}$* and *tRNA$^{Lys}$* genes. The *tRNA$^{Lys}$* gene moved from the typical neobatrachian position between *COX2* and *ATP8* genes to a

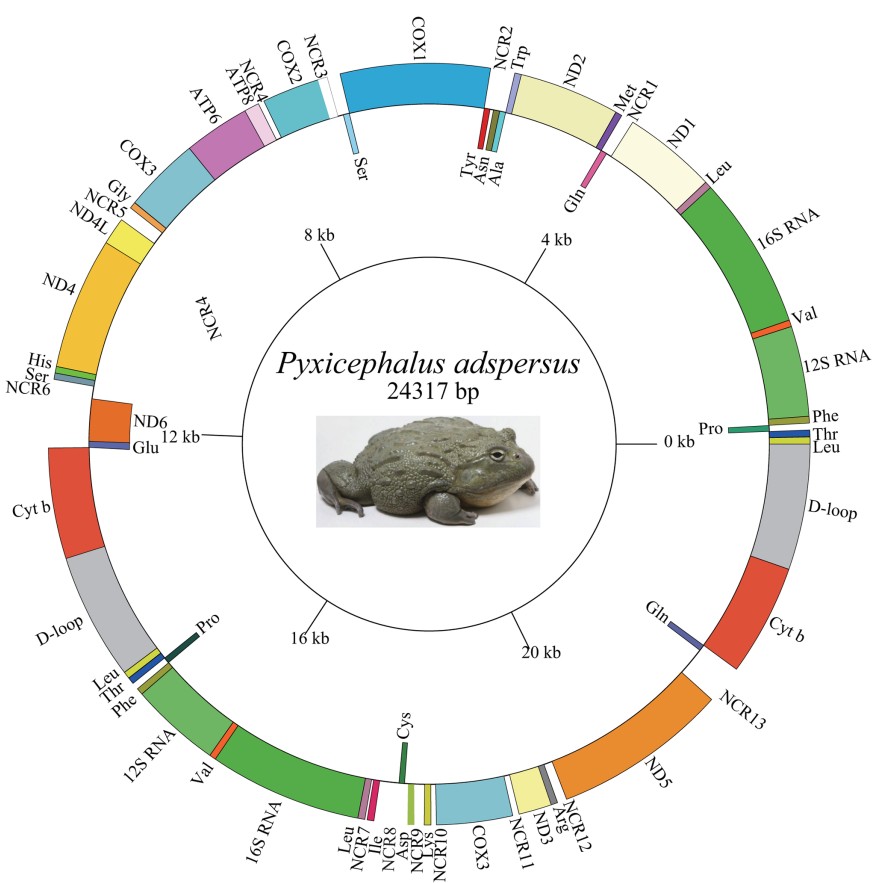

**Figure 1 Graphical map of the mitogenome of *P. adspersus*.** The tRNAs are labeled according to the three-letter amino acid codes. A gene name outside the circle indicates that the direction of transcription is anti-clockwise, whereas a gene name inside the circle indicates that the direction of transcription is clockwise.                 

location between *tRNA^Cys* and the extra *COX3* gene. The *ND3-tRNA^Arg* genes moved from the position between *tRNA^Gly* and *ND4L* genes to a location between the *COX3* and *ND5* genes. The *ND5* gene moved from the typical neobatrachian position between *tRNA^Ser* and *ND6* genes to a location between the *tRNA^Arg* and *tRNA^Glu* genes.

The mt genome of *Pyxicephalus adspersus* contained two copies of the *16S rRNA* and *12S rRNA* genes and each copy was identical to its original. The two copies of the 16S rRNA gene with a length of 1,578 bp were located between two copies of *tRNA^Leu* and *tRNA^Val* genes, respectively, whereas the two copies of the *12S rRNA* gene with a length of 932 bp were located between two copies of *tRNA^Phe* and *tRNA^Val* genes, respectively (Fig. 1). We found that the AT-skew was slightly positive whereas the GC-skew was strongly negative in the skew of the rRNA genes (Table 4), which showed that the contents of A and C were higher than those of T and G, respectively.

The mt genome of *Pyxicephalus adspersus* contained 29 tRNA genes: two copies of *LTPF* gene clusters, two copies of *tRNA^Leu*, two copies of *tRNA^Val*, two copies of *tRNA^Gln*; the tRNA sequences of each copy were identical. However, *tRNA^Ile*, *tRNA^Cys*, and *tRNA^Lys* genes were transferred from the original regions. The existence of two copies of a tRNA

Cai et al. (2019), *PeerJ*, DOI 10.7717/peerj.7532

**Table 2 Location of features in the mitogenome of *P. adspersus*.**

| Gene/region | Start position | Spacer (+) overlap (−) | Length (bp) | Start codon | Stop codon | Strand |
|---|---|---|---|---|---|---|
| tRNA$^{Leu}$ | 1–72 | +1 | 72 | | | H |
| tRNA$^{Thr}$ | 74–143 | 0 | 70 | | | H |
| tRNA$^{Pro}$ | 144–212 | +1 | 69 | | | L |
| tRNA$^{Phe}$ | 214–283 | 0 | 70 | | | H |
| 12S rRNA | 284–1,215 | 0 | 932 | | | H |
| tRNA$^{Val}$ | 1,216–1,283 | 0 | 68 | | | H |
| 16S rRNA | 1,284–2,861 | 0 | 1,578 | | | H |
| tRNA$^{Leu\ (UUR)}$ | 2,862–2,934 | 0 | 73 | | | H |
| ND1 | 2,935–3,897 | +60 | 963 | ATG | TAG | H |
| NCR1 | 3,898–3,957 | | 60 | | | |
| tRNA$^{Gln}$ | 3,958–4,028 | −1 | 71 | | | L |
| tRNA$^{Met}$ | 4,028–4,096 | 0 | 69 | | | H |
| ND2 | 4,097–5,134 | −2 | 1,038 | ATT | TAG | H |
| tRNA$^{Trp}$ | 5,133–5,201 | 0 | 69 | | | H |
| tRNA$^{Ala}$ | 5,202–5,271 | 0 | 70 | | | L |
| tRNA$^{Asn}$ | 5,272–5,344 | +39 | 73 | | | L |
| NCR2 | 5,345–5,383 | | 39 | | | |
| tRNA$^{Tyr}$ | 5,384–5,450 | +1 | 67 | | | L |
| COX1 | 5,452–6,994 | +2 | 1,543 | GTG | T | H |
| tRNA$^{Ser\ (UCN)}$ | 6,997–7,067 | +69 | 71 | | | L |
| NCR3 | 7,067–7,130 | | 59 | | | |
| COX2 | 7,131–7,818 | +61 | 688 | ATG | T | H |
| NCR4 | 7,817–7,879 | | 61 | | | |
| ATP8 | 7,880–8,044 | −10 | 165 | ATG | TAA | H |
| ATP6 | 8,035–8,716 | 0 | 682 | ATG | T | H |
| COX3 | 8,717–9,500 | +1 | 784 | ATG | T | H |
| tRNA$^{Gly}$ | 9,501–9,569 | +150 | 69 | | | H |
| NCR5 | 9,570–9,719 | | 150 | | | |
| ND4L | 9,720–10,007 | −7 | 288 | ATG | TAA | H |
| ND4 | 10,001–11,363 | 0 | 1,363 | ATG | T | H |
| tRNA$^{His}$ | 11,364–11,430 | 0 | 67 | | | H |
| tRNA$^{Ser\ (AGY)}$ | 11,431–11,497 | +143 | 67 | | | H |
| NCR6 | 11,496–11,640 | | 143 | | | |
| ND6 | 11,641–12,126 | 0 | 486 | ATG | AGG | L |
| tRNA$^{Glu}$ | 12,127–12,196 | +1 | 70 | | | L |
| Cyt b | 12,198–13,349 | 0 | 1,152 | ATG | TAA | H |
| CR1 | 13,350–14,646 | 0 | 1,297 | | | H |
| tRNA$^{Leu}$ | 14,647–14,718 | +1 | 72 | | | H |
| tRNA$^{Thr}$ | 14,720–14,789 | 0 | 70 | | | H |
| tRNA$^{Pro}$ | 14,790–14,858 | +1 | 69 | | | L |
| tRNA$^{Phe}$ | 14,860–14,929 | 0 | 70 | | | H |

| Gene/region | Start position | Spacer (+) overlap (−) | Length (bp) | Start codon | Stop codon | Strand |
|---|---|---|---|---|---|---|
| *12S rRNA* | 14,930–15,861 | 0 | 932 | | | H |
| *tRNA^Val* | 15,862–15,929 | 0 | 68 | | | H |
| *16S rRNA* | 15,930–17,507 | 0 | 1,578 | | | H |
| *tRNA^Leu (UUR)* | 17,508–17,580 | +22 | 73 | | | H |
| NCR7 | 17,579–17,602 | | 22 | | | |
| *tRNA^Ile* | 17,603–17,673 | +219 | 71 | | | H |
| NCR8 | 17,672–17,866 | | 219 | | | |
| OL | 17,867–17,895 | | 29 | | | L |
| *tRNA^Cys* | 17,893–17,957 | +9 | 65 | | | L |
| *tRNA^Asp* | 17,967–18,035 | +151 | 69 | | | H |
| NCR9 | 18,034–18,186 | | 151 | | | |
| *tRNA^Lys* | 18,187–18,256 | +54 | 70 | | | H |
| NCR10 | 18,255–18,310 | | 54 | | | |
| *COX3* | 18,311–19,094 | +65 | 784 | ATG | T | H |
| NCR11 | 19,093–19,159 | | 65 | | | |
| *ND3* | 19,160–19,501 | −2 | 342 | ATG | TAA | H |
| *tRNA^Arg* | 19,500–19,568 | +98 | 69 | | | H |
| NCR12 | 19,569–19,666 | | 98 | | | |
| *ND5* | 19,667–21,472 | +324 | 1,806 | ATG | AGG | H |
| NCR13 | 21,471–21,796 | | 324 | | | |
| *tRNA^Glu* | 21,797–21,867 | +1 | 71 | | | L |
| *CYTB* | 21,869–23,020 | 0 | 1,152 | ATG | TAA | H |
| *CR2* | 23,021–24,317 | | 1,297 | | | H |

gene is uncommon. The size of the tRNAs was 2,021 bp with an average A+T content of 56.6%. Among the 29 tRNAs, most tRNA genes excluding *tRNA^Ser (AGN)* have the common cloverleaf secondary structure. The mt genome of *Pyxicephalus adspersus* also contained two copies of the CR and each *CR* copy had an identical length of 1,306 bp and was located between two copies of the *Cyt b* and *tRNA^Leu* genes.

## Possible gene rearrangement mechanisms of *Pyxicephalus adspersus* mtDNA

In *Pyxicephalus adspersus*, two copies of the *tRNA^Glu*-Cyt b-CR-*tRNA^Leu*-*tRNA^Thr*-*tRNA^Pro*-*tRNA^Phe*-12S RNA-*tRNA^Val*-16S RNA-*tRNA^Leu* gene cluster, two copies of the *COX3* gene, and the translocation of *tRNA^Ile*, *tRNA^Cys*, *tRNA^Asp*, *tRNA^Lys*, *tRNA^Arg*, *ND3* and *ND5* genes from their typical positions were found. Comparing the mt gene rearrangements in all sequenced mitochondrial genomes of frogs, we did not find any other species where similar gene rearrangements existed. The Tandem duplication and random loss model (TDRL) (*Arndt & Smith, 1998*) can be used to explain the gene rearrangement in *Pyxicephalus adspersus*. The rearrangement mechanism of hypothesized intermediate steps is as follows (Fig. 3). Firstly, the two mitochondrial genomes were

**Table 3 The comparability between non-coding region (NCR) and the deleted corresponding genes.**

| Non-coding region | Corresponding genes | Comparability (%) |
|---|---|---|
| NCR1 | Ile | 50 |
| NCR2 | $O_L$-tRNA$^{Cys}$ | 48.7 |
| NCR3 | Asp | 59.4 |
| NCR4 | Lys | 50.7 |
| NCR5 | ND3-tRNA$^{Arg}$ | 67.3 |
| NCR6 | ND5 | 60.1 |
| NCR7 | ND1 | 100 |
| NCR8 | Gln-Met-ND2-tRNA$^{Trp}$-tRNA$^{Ala}$-tRNA$^{Asn}$ | 53.0 |
| NCR9 | COX2 | 50.3 |
| NCR10 | ATP8-ATP6 | 57.4 |
| NCR11 | Gly | 73.9 |
| NCR12 | ND4-ND4L-tRNA$^{His}$-tRNA$^{Ser}$ | 48.4 |
| NCR13 | ND6 | 92.6 |

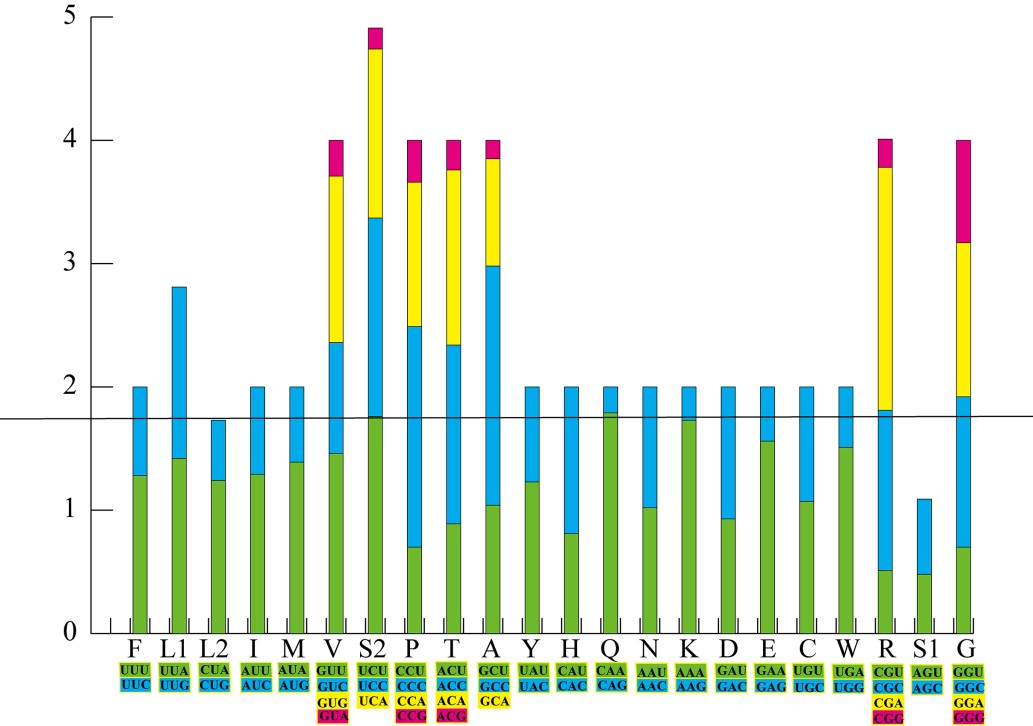

**Figure 2 The relative synonymous codon usage (RSCU) in the *P. adspersus* mitogenome.** Codon families are provided on the *x*-axis along with the different combinations of synonymous codons that code for the same amino acid. RSCU are provided on the *y*-axis.

linked by the head-to-tail method and the inferred "dimer-mitogenome" intermediate of the *Pyxicephalus adspersus* mtDNA could be formed from two entire mitogenomes. Secondly, some copies of duplicated genes were randomly deleted completely or partially from the two mt monomers. Thirdly, some duplicated genes lost their functions
**Table 4  Base composition of *P. adspersus* mitogenome.**

|  | A% | T% | G% | C% | AT% | GC% | AT skewness | GC skewness |
|---|---|---|---|---|---|---|---|---|
| Whole genome | 29.5 | 28.2 | 14.5 | 27.8 | 57.7 | 42.3 | 0.022 | −0.314 |
| Protein-coding genes | 26.1 | 30.0 | 14.3 | 29.6 | 56.1 | 43.9 | −0.071 | −0.348 |
| 1st codon positions | 26.9 | 28.1 | 18 | 27.1 | 55 | 45.1 | −0.022 | −0.202 |
| 2nd codon positions | 23.5 | 33.5 | 13.1 | 29.9 | 57 | 43 | −0.175 | −0.392 |
| 3rd codon positions | 27.9 | 28.5 | 11.9 | 31.7 | 56.4 | 43.6 | −0.012 | −0.455 |
| tRNA genes | 28.8 | 27.8 | 22.1 | 21.2 | 56.6 | 43.3 | 0.018 | 0.021 |
| rRNA genes | 33.3 | 24.5 | 18.2 | 24.0 | 57.8 | 42.2 | 0.151 | −0.139 |
| Control region | 31.9 | 33.6 | 12.5 | 22.0 | 65.5 | 34.5 | −0.026 | −0.275 |

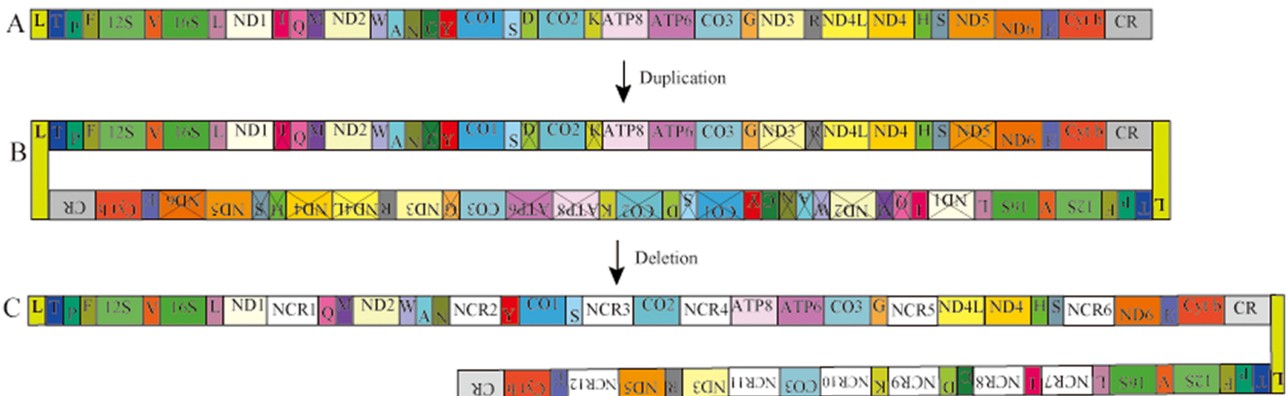

**Figure 3  Proposed mechanism of gene rearrangements in the *P. adspersus* mitogenome under the Dimer-Mitogenome and a model of tandem duplication of gene regions.** (A) Typical Dicroglossidae gene order. (B) The dimeric molecule with two monomers linked head-to-tail and subsequent first deletions of partial genes or complete genes resulting in the derived gene order. (C) State in *P. adspersus* after the final duplication and deletion.

or became noncoding regions. This then formed the gene arrangement of *Pyxicephalus adspersus*. Hence, the Dimer-Mitogenome and the TDRL model (*Arndt & Smith, 1998*) may be the most appropriate to explain the gene arrangements in *Pyxicephalus adspersus*.

## Phylogenetic analysis

We illustrated nodal supports from ML and BI analyses together using Bayesian topology (Fig. 4) because the combined data set consisting of 10 protein coding sequences resulted in identical topology for phylogenetic relationships. Both ML and BI analyses showed high branch support values. Dicroglossidae is a sister clade of Ranidae. The clade of Mantellidae and Rhacophoridae is a sister clade of Dicroglossidae and Ranidae. Petropedetidae is a sister clade of Ptychadenidae and then Pyxicephalidae is a sister clade of (Petropedetidae + Ptychadenidae). The clade of (Pyxicephalidae + (Petropedetidae + Ptychadenidae)) is a sister clade of (Dicroglossidae + Ranidae) + (Mantellidae + Rhacophoridae). Ceratobatrachidae is a sister clade of Phrynobatrachidae. The clade of

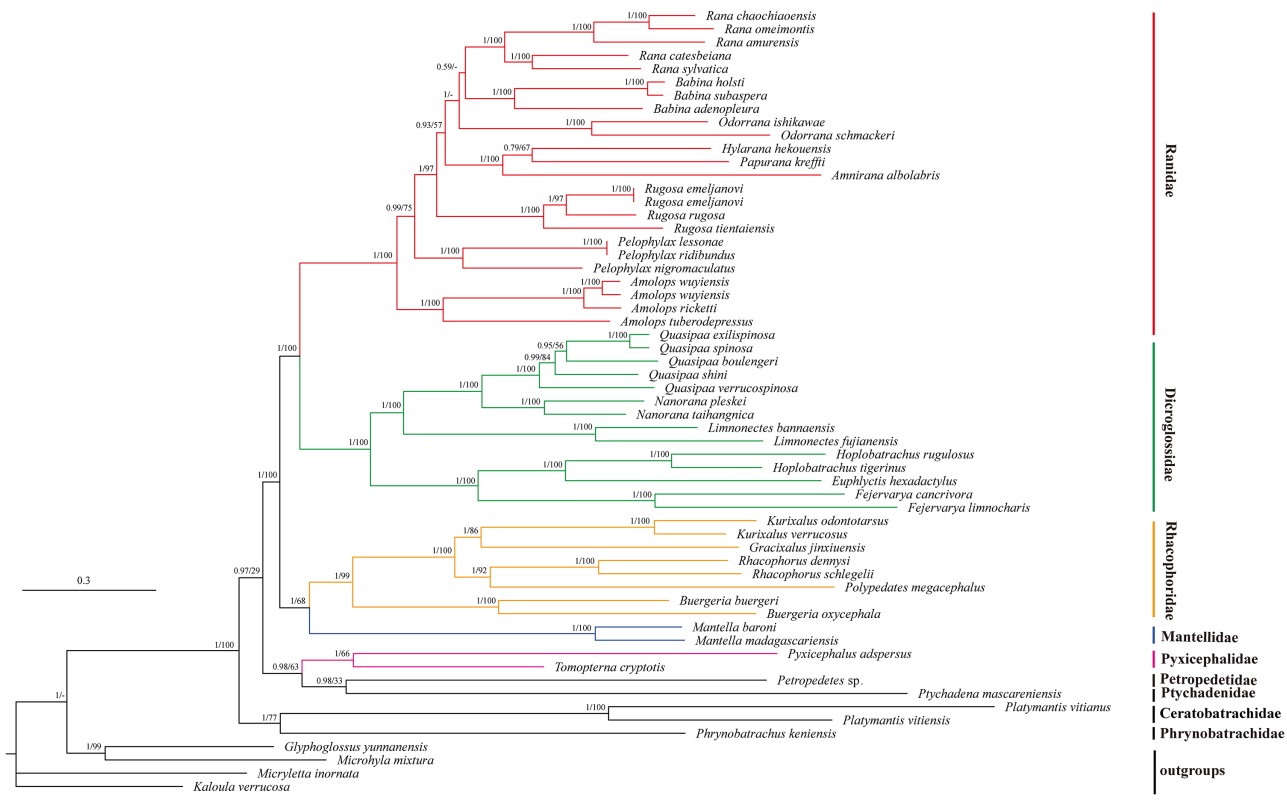

**Figure 4 Phylogenetic relationships of Ceratobatrachidae, Dicroglossidae, Mantellidae, Petropedetidae, Phrynobatrachidae, Pyxicephalidae, Ptychadenidae, Ranidae, and Rhacophoridae based on 10 protein-coding genes using nucleotide datasets.** Phylogenetic analyses using nucleotide datasets were carried out for 59 frog species based on all 10 protein-coding genes from their respective mt genomes. Branch lengths and topology are from BI analysis. The tree was rooted with four out-groups (*Micryletta inornata*, *M. mixtura*, *Glyphoglossus yunnanensis*, and *Kaloula verrucosa*). Numbers above the nodes are the posterior probabilities of BI on the left and the bootstrap values of ML on the right.

Ceratobatrachidae and Phrynobatrachidae is a sister clade of ((Pyxicephalidae + (Petropedetidae + Ptychadenidae)) + (Dicroglossidae + Ranidae) + (Mantellidae + Rhacophoridae). *Pyxicephalus adspersus* is a sister clade of *Tomopterna cryptotis*. The monophyly of Dicroglossidae, Ranidae, Mantellidae, Rhacophoridae, Pyxicephalidae, and Ceratobatrachidae was well supported.

The monophyly of Pyxicephalidae is supported for *Pyxicephalus adspersus* (Pyxicephalinae) as a sister clade of *T. cryptotis* (Cacosterninae). Pyxicephalidae is a valid family in this study as was also supported by the results of *Frost et al. (2006)*, *Roelants et al. (2007)*, and *Pyron & Wiens (2011)*. Although Scott and Van Der Meijden supported Pyxicephalinae as imbedded within the Discroglossine (*Scott, 2005*; *Van Der Meijden et al., 2005*), *Pyxicephalus adspersus* (Pyxicephalidae) is not clustered into Dicroglossidae or Ranidae, whereas Dicroglossidae is a sister clade of Ranidae, which is supported by *Zhang et al. (2018a)*. *Zhang et al. (2013)* supported Pyxicephalidae as a sister clade to Petropedetidae, our results supported Pyxicephalidae as monophyletic and Pyxicephalidae was a sister clade of Petropedetidae and Ptychadenidae.

## CONCLUSIONS

We successfully determined the complete mitogenome of *Pyxicephalus adspersus*. The length of the *Pyxicephalus adspersus* mitogenome (24,317 bp) was the largest size among all known anurans mitogenomes. We found two copies of the *tRNA$^{Glu}$-Cyt b-CR-tRNA$^{Leu}$-tRNA$^{Thr}$-tRNA$^{Pro}$-tRNA$^{Phe}$-12S RNA-tRNA$^{Val}$-16S RNA-tRNA$^{Leu}$* gene clusters, two copies of *COX3* genes, and the translocation of *tRNA$^{Ile}$, tRNA$^{Cys}$, tRNA$^{Asp}$, tRNA$^{Lys}$, tRNA$^{Arg}$, ND3,* and *ND5* genes from their typical positions. The Dimer-Mitogenome and TDRL models may be the most appropriate to explain the gene arrangements in *Pyxicephalus adspersus*. In this study, both BI and ML analyses supported the conclusion that Pyxicephalidae was monophyletic and Pyxicephalidae was the sister clade of (Petropedetidae + Ptychadenidae).

## ACKNOWLEDGEMENTS

We are grateful to Le-Ping Zhang for her help in the study.

### Funding

This research was supported by the Natural Science Foundation of Zhejiang Province (LQ16C030001), and the National Natural Science Foundation of China (Nos. 31801963) for the study design, data collection and analyses. The funders had no role in study design, data collection and analysis, decision to publish, or preparation of the manuscript.

### Grant Disclosures

The following grant information was disclosed by the authors:
Natural Science Foundation of Zhejiang Province: LQ16C030001.
National Natural Science Foundation of China: 31801963.

### Competing Interests

Kenneth B. Storey is an Academic Editor for PeerJ. Jia-Yong Zhang is an Academic Editor for PeerJ.

### Author Contributions

- Yin-Yin Cai conceived and designed the experiments, performed the experiments, analyzed the data, prepared figures and/or tables, authored or reviewed drafts of the paper.
- Shi-Qi Shen conceived and designed the experiments, performed the experiments, prepared figures and/or tables, authored or reviewed drafts of the paper.
- Li-Xu Lu performed the experiments, prepared figures and/or tables, authored or reviewed drafts of the paper.
- Kenneth B. Storey prepared figures and/or tables, authored or reviewed drafts of the paper.
- Dan-Na Yu conceived and designed the experiments, performed the experiments, analyzed the data, contributed reagents/materials/analysis tools, prepared figures and/or tables, authored or reviewed drafts of the paper.
• Jia-Yong Zhang conceived and designed the experiments, performed the experiments, contributed reagents/materials/analysis tools, prepared figures and/or tables, authored or reviewed drafts of the paper, approved the final draft.

## Animal Ethics

The following information was supplied relating to ethical approvals (i.e., approving body and any reference numbers):

Sample acquisition was reviewed, approved and carried out in accordance with the relevant guidelines of the Committee of Animal Research Ethics of Zhejiang Normal University.

## DNA Deposition

The following information was supplied regarding the deposition of DNA sequences:

The mt genome described here is accessible via GenBank accession number MK460224.

## Data Availability

The raw measurements are available as a Supplemental File.

## Supplemental Information

Supplemental information for this article can be found online at http://dx.doi.org/10.7717/peerj.7532#supplemental-information.

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
