# Peer review of "The complete mitochondrial genome of Pyxicephalus adspersus: high gene rearrangement and phylogenetics of one of the world’s largest frogs"

_PeerJ, doi:10.7717/peerj.7532_

## Round 0.1 · original submission · Major Revisions

I received two reviews about your manuscript. Both reviewers recognize the interesting results presented in your manuscript, but they suggest that the text necessitates important ameliorations in terms of form. Reviewer 1 insists on the written English too, and I have myself believe that the text needs some rewriting and polishing. I also noted that there are very large and unecessary lists of references that should be limited to a few significant examples. The interpretation of the results in the phylogenetic part is also particularly needing revision as presented by the first reviewer.

Reviewer 1 ·

Basic reporting

Literature references, sufficient field background/context provided.
It needs more detail in the introduction to explain the value of studying Pyxicephalus adspersus, especially its complete mitogenome.

Experimental design

The complete mt genome results showed that there are 15 PCGs for Pyxicephalus adspersus. Why 10 PCGs were chosen as an concatenated dataset to implement phylogenetic analyses, and what protein coding genes were chosen? Why not choose 13 PCGs or 15 PCGs? In addition, the authors named the concatenated dataset as 10P123 in line 167, while it was changed as 13P123 in line 168. The best substitution models obtained from PartitionFinder in line 174 were not the same as the authors used in BI and ML analyses, please explain why.

Validity of the findings

1. Some academic terminology abbreviations are not homologous. For example, the expression “the trnT gene” was used in line 80, while “tRNAs Ile, Cys, Lys, and Arg” was found in line 118.
2. There are some inappropriate sentences in the manuscript, e.g., in line 226 “…located between ND6 and Ile genes” should be “… ND6 and tRNAIle genes”. The same expressions were found in line 228, 229, 230, etc.
3. Only one species, Pyxicephalus adspersus, representing the family Pyxicephalinae cannot explain its monophyly.

Additional comments

The English language should be improved to ensure that your international audience can clearly understand your text. I suggest a pre-review of your manuscript from your native English speaking colleague before your submission. Some examples where the language could be improved, e.g. lines 51, 96, 170, 171 etc - the current phrasing makes comprehension difficult. In the whole manuscript, it is very interesting about the rearrangement of mitochondrial genome, while the authors did not address this in phylogenetic analyses, and the monophyly of family Pyxicephalinae was not well supported in this study. I suggest the major revision before resubmission.

Reviewer 2 ·

Basic reporting

Some sentence are unclear, and some abbreviation are not normalized.
1. Line 47 It's an incomplete sentence. “and” what?
2. Line 63 It’s better to add a topic sentence in this paragraph, as the author described many kinds of gene rearrangement in anurans.
3. Line 164 The genes name should be normalized and unified in one paper.
4. Line 168 what is “13P123”? Is there any difference between “10P123”?
For 10 mt PCGs, it’s no need to describe the name “10P123”.
5. Line 187 This sentence is ambiguous.
6. Line 193 What is “NCR12”? The definition should be shown for the first time of the paper.

Experimental design

no comment

Validity of the findings

The manuscript by Cai et al. is an important contribution regarding mitochondrial genome of Pyxicephalus adspersus and the anurans mt evolution. The results of NCR need check.
Line 208-209 There should be another NCR between Arg and ND5.

Additional comments

The author sequenced the whole mt genome of this species, and found specific gene rearrangement. Now, the genome of P. adspersus has been sequenced, but the mt genome of this species is lacking. The results of this study could be used to identify the mt sequences and helpful understanding the phylogeny and mt evolution of anurans.

---

## Round 0.2 · Minor Revisions

I have read your letter and the modified manuscript. Thank you for taking most of the suggestions of the reviewers into account. One of the previous reviewers agreed to evaluate again your manuscript and also noticed important amelioration. However he makes a few additional suggestions and points out some changes that still need to be addressed before paper acceptance.

Reviewer 1 ·

Basic reporting

1. The sentence in Line 91 should be deleted

2. Line 103, “Pyxicephalinae as one subfamily of Ranidea” or Ranoidea?

3. Line 97, it’s better to add a topic sentence in this paragraph, as the author described the taxonomic status of Pyxicephalidea.

4. Line 120, the sentence is not precise or persuasive. “a species with a giant size with an animal group may be associated with a larger mt genome” may be reasonable in frogs but not in all other species. Although the following example was given to support this sentence, please give more references.

Experimental design

1. Line 196, can be modified as:
Although extra COX3 and Cyt b genes were found in Pyxicephalus adspersus (see the result in the following context), the extra copies of COX3 and Cyt b genes were identical to the other COX3 and Cyt b genes (100% similarity). Therefore, in phylogenetic analyses, only one set genes of COX3 and Cytb was used. In addition, due to some mitochondrial protein-coding genes missing in some species (ND5), lacking good information (ATP8) and the heterogeneous base composition and poor phylogenetic performance (ND6) (Zhang et al., 2018a), ten PCGs genes were used in this study and separately aligned in Mega 7.0 (Stecher et al., 2016).

2. The best substitution models for BI and ML can be both obtained from PartitionFinder. The models of each partition can be set separately. It is not reasonable to replace GTRGAMMAI model in RaxML and GTR+I+G in BI. Please explain why and do the new analyses.

Validity of the findings

1. Line 246, Ile should be tRNAIle gene?

2. The discussion about the monophyly of Pyxicephalidae should be in the discussion or conclusion part. In addition, the monophyly of Pyxicephalidae should be confirmed by more adequate sampling in future studies.

Additional comments

The authors have amended the manuscript very quickly and carefully and given positive feedback in most of the reviewing points. But there are still some sentences should be modified. In addition, it is better to do a new phylogenetic analyze to confirm the results.

---

## Round 0.3 · accepted · Accept

Thank you for the revised version of your manuscript and the replies to the reviewer. Your manuscript is now ready for publication.